# H pilin cyclisation and pilus biogenesis are promiscuous but electrostatic perturbations impair conjugation efficiency

Shan He [1,6], Naito Ishimoto [1,2,5,6], Joshua L. C. Wong [1,6], Sophia David [3], Julia Sanchez-Garrido [1], Mikhail Bogdanov [4], Konstantinos Beis [1,2] ✉ & Gad Frankel [1] ✉

During conjugation, plasmid DNA is transferred from donor to recipient bacteria via the plasmid-encoded mating pilus, formed as thin helical assemblies of polymerised pilin subunits. In the IncHI1 R27 plasmid-encoded pilus, the TrhA pilin undergoes cyclisation (via a peptide bond between Gly1 and Asp69), essential for conjugation. Gly1 and Asp69 are exposed on the pilus surface and conserved in all TrhA pilins in the Plascad database. Substituting Asp69 with Asn, Ala, Gly, or Arg does not prevent cyclisation or pilus formation, which remains structurally indistinguishable from the wild type. Conjugation efficiency of the Asp69 substitutions across multiple recipient species correlates with side chain size, in the order Asp69Asn > Asp69Ala > Asp69Gly. However, Asp69Arg, as well as Asp69Lys and Gly1Lys substitutions abolish conjugation, likely due to the positively charged pilus surface (opposite to the wild-type negative charge) forming unfavourable electrostatic interactions with the recipient outer membrane's inner leaflet, composed solely of zwitterionic phosphatidylethanolamine (PE). Consistently, conjugation is rescued in recipients lacking PE. These findings indicate strong selective pressure to maintain Gly1 and Asp69, as efficient DNA transfer depends on precise electrostatic and steric constraints of the pilus surface.

Bacterial evolution is driven by random mutations and horizontal gene transfer (HGT) followed by natural selection. HGT is mediated by transformation, transduction and conjugation[1,2]. Conjugation, first discovered in 1946 using the F plasmid[3], facilitates the acquisition and dissemination of metabolic, virulence and antibiotic resistance genes (ARGs)[4]. Conjugation takes place in a broad range of environments, including soil, aquatic ecosystem and the lumen of the gut. In contrast to transformation and transduction, during conjugation DNA is transferred unidirectionally from one bacterium (donor) to another (recipient) in a contact-dependent manner[5,6].

Conjugative plasmids are found in both Gram-negative and Gram-positive bacteria[5]. Multiple classification systems have been used to categorise different plasmid-encoded conjugation systems, including a relaxase phylogenetic diversity, which defines nine mobility (MOB) groups[7], and mating pair formation (MPF) classification which defines four groups: MPFF, MPFG, MPFI and MPFT[8]. Plasmids in the MPFI

[1]Department of Life Sciences, Imperial College London, London, UK. [2]Rutherford Appleton Laboratory, Research Complex at Harwell, Didcot Oxfordshire, UK. [3]Centre for Genomic Pathogen Surveillance, Pandemic Sciences Institute, University of Oxford, Oxford, UK. [4]Department of Biochemistry & Molecular Biology, The University of Texas Health Science Center, McGovern Medical School, Houston, TX, USA. [5]Present address: Drug Design Laboratory, Graduate School of Medical Life Science, Yokohama City University, Tsurumi, Yokohama, Japan. [6]These authors contributed equally: Shan He, Naito Ishimoto, Joshua L. C. Wong. ✉e-mail: kbeis@imperial.ac.uk; gad.frankel@imperial.ac.uk

group belong to the incompatibility (Inc) groups IncI and IncL/M, plasmids in the MPFT group belong to IncN, IncP and IncW, and those in the MPFF group belong to IncA, IncC, IncF and IncH.

The MPFF conjugation process is divided into three phases[8,9]. Phase 1 occurs exclusively within the donor, which assembles the type IV secretion system injectisome[10] and the mating pili, built as a thin helical pilin assembly with phospholipid molecules[11–13]. Phase 2 involves both the donor and recipient, where initial contacts are mediated by the pilus in the donor and an unknown receptor on the recipient, a process called mating pair formation (MPF)[5,6,14]. During MPF plasmid DNA is mobilised from the donor to the recipient in low efficiency[5,15]. MPF-mediated contact is believed to induce retraction of the pilus, bringing the donor and recipient into proximity through a process called mating pair stabilisation (MPS)[5,16]. MPS-mediated intimate contact enables efficient plasmid transfer[14]. Phase 3 occurs exclusively within the recipient, which following plasmid entry starts to express plasmid genes involved in anti-defence systems and exclusion[17,18].

Recently, using a de-repress and chloramphenicol resistant derivative of the reference IncHI plasmid, R27 (drR27), we determined the cryo-EM structure of the H-pilus at 2.2 Å resolution[12]. Like the pOX38/pED208/pKpQIL (IncF)[11–13], R388 (IncW)[19] and pKM101 (IncN)[20], the H-pilin subunits, TrhA, form helical assemblies with negatively charged phosphatidylglycerol (PG) molecules at a stoichiometric ratio of 1:1[12]. We found that unlike all other lipidated pilin structures, which adopt a hairpin topology, TrhA forms a cyclic structure. We have shown that cyclisation is mediated following cleavage of the N-terminal signal peptide and the five C-terminal residues of the pre-pilin. Cyclisation occurs via condensation of the amine and carboxyl residues on Gly1 (G1) and Asp69 (D69) to generate the mature pilin. Conjugation assays and negative staining EM revealed that cyclisation is essential for biogenesis of a functional pilus, as the non-cyclic TrhA$_{\Delta 67-74}$ did not assemble into a pilus[12]. Previous biochemical studies have shown that the RP4 plasmid (IncP) pilin TrbC also forms a cyclic structure[21]. We have recently confirmed the cyclic nature of TrbC by cryo-EM[22]. Cyclisation of TrbC occurs via formation of a peptide bond between Ser1 (S1) and Gly78 (G78) of the mature pilin[21,22]. Substitutions S1A/G/T or G114A/S did not affect TrbC cyclisation while TrbC G114C/L/T did not cyclise, resulting in loss of pilus biogenesis and DNA transfer[23].

In this work, we study the roles of G1 and D69 in mediating TrhA cyclisation and function, complementing our previous work probing the effect of loss of cyclisation on pilus biogenesis. Comparing the available 147 TrhA sequences in the Plascad database reveals that G1 and D69 are 100% conserved. We construct an N-terminal site directed pilin mutant, G1K, and five C-terminal site directed pilin mutants, D69N, D69A, D69G, D69K and D69R. We show that all site directed mutants produce a pilus. Pili made by the TrhA D69N, D69A, D69G substitutions show a trend towards a gradient of conjugation efficiency into some recipients, correlated with the amino acid side chain size, N > A > G, e.g. larger side chains generally resulting in higher efficiency. While the structures of the pili produced by D69N, D69A, D69G and D69R are indistinguishable from the wildtype pilus, the TrhA G1K, D69K and D69R substitutions abolish conjugation. However, conjugation of R27$_{G1K}$, R27$_{D69K}$ and R27$_{D69R}$ is restored using a mutant recipient unable to produce phosphatidylethanolamine (PE), the primary and only zwitterionic phospholipid in the inner (periplasmic) leaflet of the outer membrane in Gram-negative bacteria[24]. Together, these data show that while pilin cyclisation and pilus biogenesis is promiscuous, the size and charge of the distal amino acids play a significant role in conjunction efficiency.

## Results
### Distribution of TrhA variants amongst the MPFF plasmids
We assessed the sequence diversity of TrhA proteins among 148 plasmids carrying IncH replicons from the Plascad database[25]. Some of these plasmids represent "fusion" plasmids in which IncH replicons were found together with replicons from different incompatibility groups (e.g. IncF). Using a tBLASTn approach with TrhA from R27 as the query sequence, we identified a single copy of TrhA in each of the plasmids, except for one plasmid in which the gene was found fragmented (accession NZ_CP021209.1). The 147 intact TrhA proteins had a minimum pairwise sequence identity of 55.1%, with 29 positions showing variation within the 69-residue mature sequence. We found that the Gly1 and Asp69 residues within the mature sequence were fully conserved across all 147 proteins.

We next used the alignment to construct a phylogenetic tree of the 147 TrhA protein sequences (Fig. 1A; https://microreact.org/project/trhA). The tree shows that TrhA sequences are divided into three main homology groupings, each of which are associated with distinct IncH replicon subtypes. The first group, which contains 24 TrhA(R27) sequences, are largely from IncHI1A/IncHI1B(R27) plasmids found among *Escherichia coli* and *Salmonella enterica*. All but one of these sequences possess ≥98% protein identity to TrhA(R27) while the remaining sequence shares 93.2% identity. The second group contains 87 TrhA sequences (84 of which are identical) from IncHI2/IncHI2A plasmids that are associated with diverse Enterobacteriaceae species and have 65.8-70.1% protein similarity with TrhA(R27). The final group contains 36 identical TrhA sequences that are almost exclusively from *Klebsiella pneumoniae* plasmids with IncHI1B (and IncFIB(Mar)) replicons and which share ~55.1% protein similarity with TrhA(R27).

### H pilus cyclisation is promiscuous
As G1 and D69 are 100% conserved amongst the 147 TrhA pilins, we hypothesised that any substitutions would interfere with cyclisation. To test this, we first engineered four site directed TrhA C-terminal substitution mutants: D69N, D69A, D69G and D69R. The pili were purified and cryo-EM analyses confirmed that similarly to the wild type (WT) TrhA, all mutant pilins cyclised and assembled into pili indistinguishable from the WT pilus structure, with the inner diameter, 22 Å (Fig. 2).

The D69 substitutions did not alter the pilin:lipid ratio, which was maintained at 1:1, or the conformation of the bound lipid (Fig. 2). Electrostatic analysis revealed that loss of the negative charge from the D69N/A/G substitutions specifically alters the electrostatic potential of the pilus surface but did not affect the lumen. Similarly, the D69R substitution did not alter the overall lumen electrostatic potential, but it introduces a strong positive charge on the pilus exterior (Fig. 3). Mapping the side chains onto the pilus structure, it is apparent that the side chain of D69 projects away from the surface, explaining why the mutants do not alter the electrostatic potential of the lumen (Fig. 4 and Supplementary Fig. 1).

### Carboxy terminal amino acid charge and side chain size affect conjugation efficiency
To determine the functional consequences of the amino acid substitutions, we performed conjugation assays using an auxotrophic *E. coli* MG1655 Δ*trp* harbouring WT drR27 (R27$_{WT}$), R27$_{D69N}$, R27$_{D69A}$, R27$_{D69G}$ or R27$_{D69R}$ as donors and *E. coli* MG1655[12], enteropathogenic *E. coli* (EPEC O127:H7, E2348/69)[26], *Klebsiella pneumoniae* (ICC8001)[27,28], *Enterobacter cloacae* (ATCC 13047)[28] and *Citrobacter amalonaticus*[29] as recipients, as described[12]. As expression of the conjugation operon in IncH plasmid is thermoregulated[30], we performed conjugation assays at 25 °C.

The conjugation efficiency of R27$_{D69N}$, R27$_{D69A}$ and R27$_{D69G}$ into *E. coli* MG1655, EPEC and into *K. pneumoniae* was equivalent. However, conjugation efficiency into *E. cloacae* and *C. amalonaticus* recipients showed a trend towards direct correlation between the side chain size (i.e. the bigger the side chain the greater the conjugation efficiently) (Fig. 5A-E, Table 1).

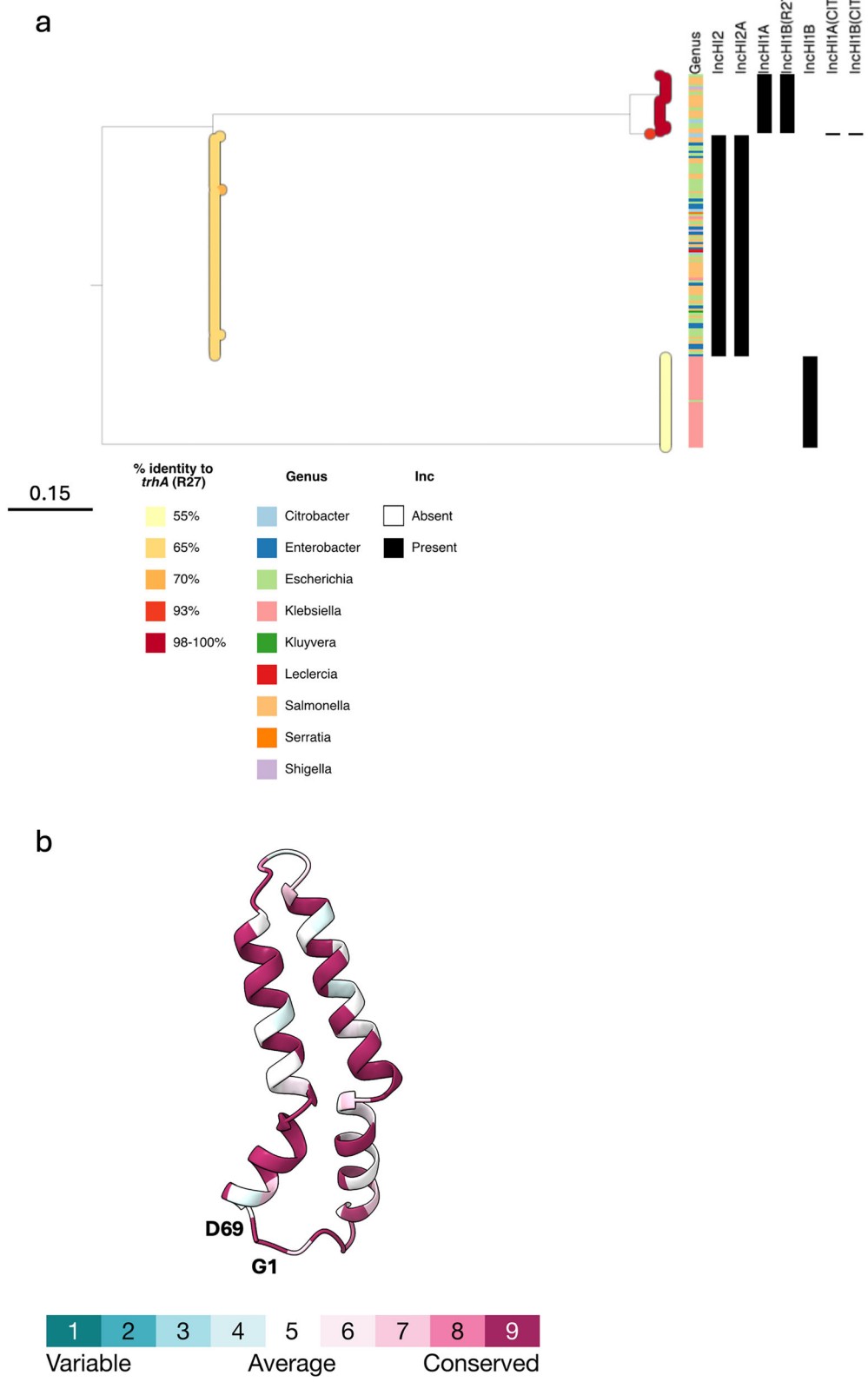

**Fig. 1 | Phylogenetic analysis of TrhA. a** Midpoint-rooted phylogenetic tree of 147 TrhA homologues from IncH plasmids. The tree tips show the percent identity of each TrhA sequence with respect to TrhA(R27). Metadata columns show the host genus of the associated plasmid and the presence/absence of matches to individual IncH replicons. The scale bar shows the number of substitutions per site. A similar visualisation can be accessed via Microreact: https://microreact.org/project/trha. **b** Conservation of the TrhA pilin amino acids from the phylogenetic tree were mapped onto the cryo-EM structure (PDB ID: 9HVC). There is a high degree of conservation amongst the different pilins. G1 and D69 are indicated.

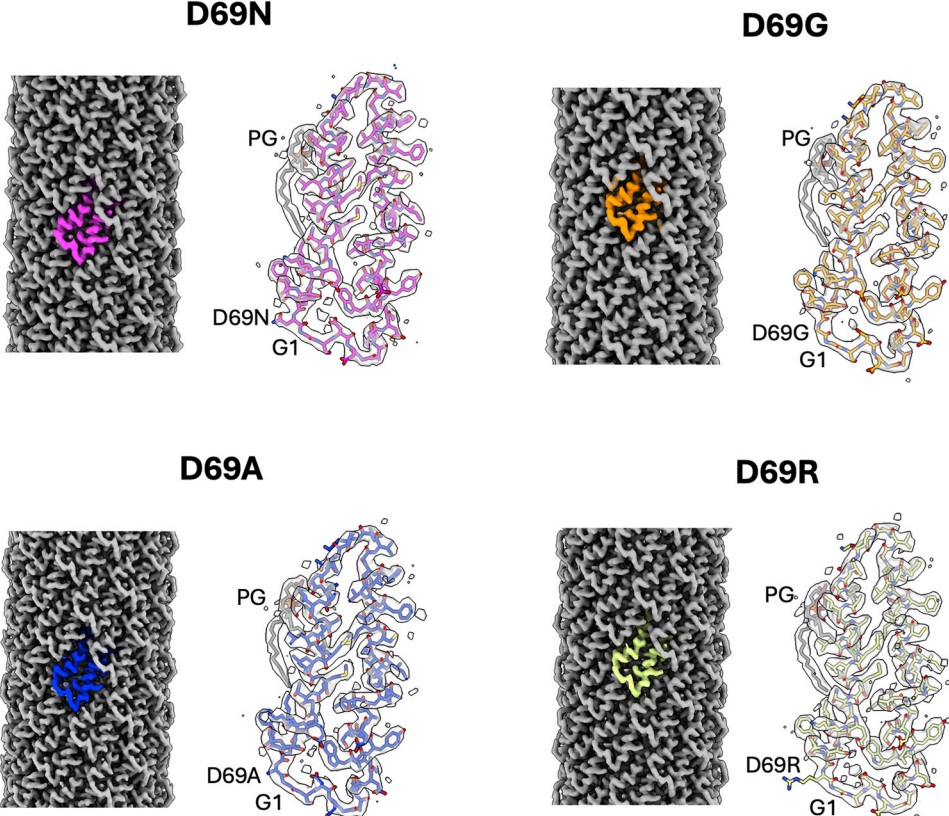

**Fig. 2 | Cryo-EM structure of the D69 N/A/G/R TrhA substitutions.** The four mutants can cyclise and assemble into a pilus. The cryo-EM maps of the reconstructed pilus are shown in grey with a single pilin subunit shown in different colours for each of the mutants. The maps unambiguously show that the mutants can cyclise. A TrhA mutant pilin subunit (shown in sticks) is placed in the map and the cyclization residues labelled. Each pilin is associated with one PG lipid (shown in grey sticks).

In contrast, the frequency of R27$_{D69R}$ conjugation was significantly lower than R27$_{WT}$ into all recipients (Fig. 5a–e, Table 2). We hypothesise that the positively charged R27$_{D69R}$ pilus surface forms electrostatic interactions with the positively charged primary amine group of PE in the outer membrane of the recipient, which disturb placing the tip into a conjugation permissive position. Alternatively, this effect could result due to the large arginine sidechain. To test this, we used an *E. coli* recipient, AL95, in which the *pssA* gene encoding phosphatidylserine synthase is interrupted by a KanR cassette[31]. This mutant remains viable but is unable to synthesize zwitterionic glycerophospholipid phosphatidylethanolamine (PE) (Fig. 5f). We also generated an R27$_{D69K}$, which maintains the carboxy terminal positive charge in the context of a small sidechain. Using donors containing R27, R27$_{D69R}$ or R27$_{D69K}$ in conjugation assays revealed that all three plasmids were conjugated in similar efficiency into the PE-deficient AL95 recipient, while R27$_{D69R}$ and R27$_{D69K}$ were conjugated in low efficiency into the WT *E. coli* recipient control, which contains ample amount of zwitterionic PE (Fig. 5g, Table 2). These results suggest that electrostatic interactions between the positively charged pilus and the positively charged primary amine group of zwitterionic PE in the outer membrane of the recipient interfere with conjugation efficiency and electrostatic constraints of the pilus surface and outer membrane of recipient cells were coevolved to ensure efficient conjugation.

### G1K substitution abolishes conjugation in WT *E. coli* recipient

As G1 is also surface exposed and 100% conserved (Fig. 1), we next investigated if a G1K substitution would affect the pilus function similarly to D69R or D69K substitutions. The SignalP 6 server predicted efficient signal peptide cleavage of the G1K pilin[32]. Functional

assays showed that, similarly to R27$_{D69K}$ and R27$_{D68R}$, R27$_{G1K}$ was conjugated in low efficiency into the WT *E. coli* recipient, but at similar efficiency as R27$_{WT}$ into AL95 (Fig. 5G, Table 2).

Taken together, these results suggest that reversing the pilus charge by chanting either the N or C terminal residue is detrimental for conjugation. Moreover, this shows that the biochemical properties of the outer membrane in the recipient might affect the spread of antimicrobial resistance genes.

## Discussion

While most conjugation pilus structures showed that the pilin has a hairpin topology, we recently reported that TrhA encoded by R27 and TrbC encoded by RP4 plasmids are cyclic[12,21,22]. Importantly, while TrbC G114A/S pilins cyclised and mediated conjugation as efficiently as wild type TrbC, TrbC G114C/L/T did not cyclise or polymerise into a pilus[20]. As such, it is not known if the side chain size rule, we observed in TrhA, is also applicable to TrbC.

Our data suggest that cyclisation of the TrhA pilin is a promiscuous process, similarly to that shown for the cyclic RP4 TrbC pilin[21]. As cyclisation of TrhA involving cleaving the five carboxy terminal amino acids and bringing the N- and C- termini of the mature pilin into proximity, the promiscuous nature of the process implies low specificity of protease/cyclase involves in recognition of N- and C-binding sites in TrhA. While TrbF is the protease/cyclase involves cyclisation of TrbC[23], the identity of the protease/cyclase involved in cyclisation of TrhA is not yet known.

Analysis of 147 TrhA proteins from IncH plasmids has shown that while they are divided into three distinct sequence homology groups, G1 and D69 are fully conserved. This suggests that selective evolutionary pressure maintains these residues.

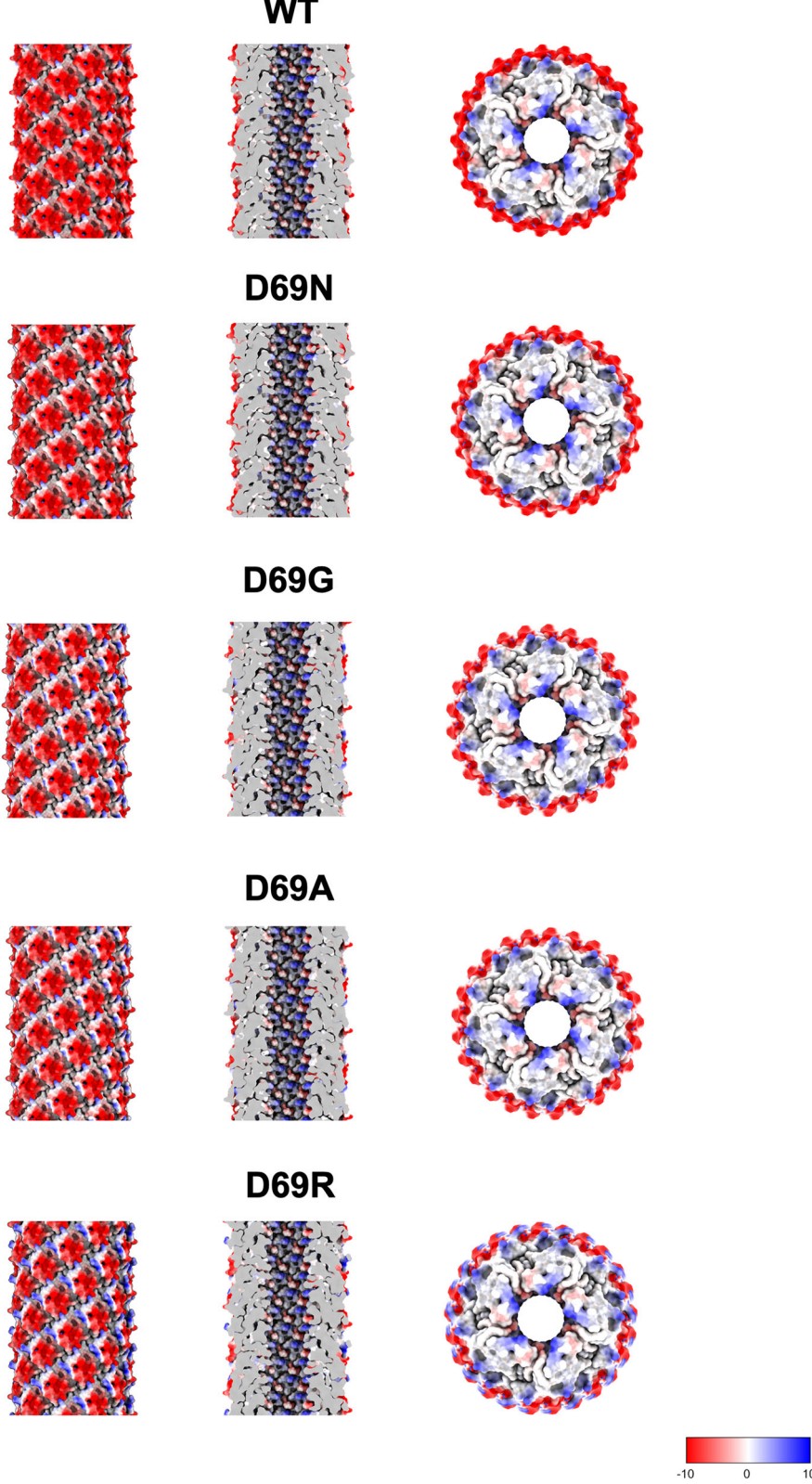

**Fig. 3 | Electrostatic potential of the TrhA D69 N/A/G/R TrhA substitutions.** The substitutions alter the charge profile of the pilus surface (left panel) but not its lumen (middle panel). The D69R results in a strong positive charge between pilin subunits. A top view (right panel) highlights how loss of the negative or gain of a positive charge specifically impacts the exterior of the pilus. The coulombic electrostatic potential is displayed as red and blue isosurfaces at levels −10 and +10, respectively; the potential is in units of kcal/(mol·$e$) at 298 K.

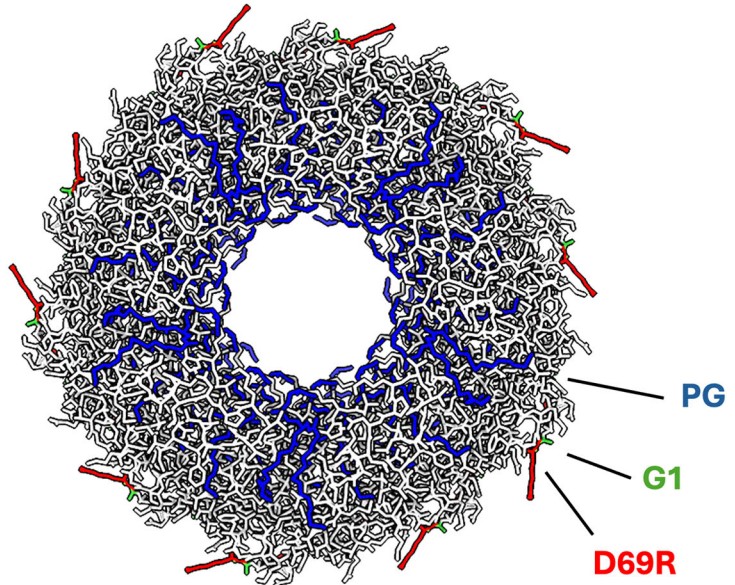

**Fig. 4 | The side chain of the cyclised C-terminal residue points on the pilus surface.** Upon cyclisation, the side chain of D69 in both the (**a**) TrhA$_{D69}$ and (**b**) TrhA$_{D69R}$, is pointing on the surface of the pilus that significantly changes the surface electrostatic potential but not of the lumen (see Fig. 3). The PG lipid (shown in blue) headgroup is pointing in the interior. The TrhA pilin is shown in grey. The key residues of a single TrhA pilin are labelled.

While the prevailing model suggests that the lipid moiety is essential for pili biogenesis and for repelling the negatively charged DNA during transfer of the relaxase-ssDNA complex[11,33,34], we have recently reported that the RP4 pilus, made by a cyclic TrbC pilin in the absence of lipid, is functional[22]. This raises the possibility that coating the lumen in negative charge might not be essential for conjugation. Indeed, in the case of the T pilus, encoded by the Ti plasmids of *Agrobacterium tumefaciens*, the lumen is coated by a positive charge due to the presence of zwitterionic PE phospholipid[21,23,35]. In fact, here we report that the exterior charge of the pilus affects conjugation, as the TrhA G1K, D69R and D69K substitutions, which resulted in pilus biogenesis, mediated conjugation at frequency of ca $10^6$ lower than R27$_{WT}$.

R27 pilus is primarily composed of the TrhA pilin subunits. As D69 is surface exposed we assessed the functional consequences of neutral R27$_{D69N}$, R27$_{D69A}$ and R27$_{D69G}$, as well as polar R27$_{D69R}$ and R27$_{D69K}$ substitutions. This revealed that the neutral animo acids substitution did not affect conjugation efficiency into *E. coli* and *K. pneumoniae* recipients. In contrast, we recorded a direct correlation between the side chain size and conjugation efficiency into *E. cloacae* and *C. amalonaticus*, with bigger chide chains mediating greater conjugation efficiency. This might reflect differences in the composition and properties of the outer membrane in the recipients.

With regards to the polar substitutions, we show that K1 and R69 projects further from the pilus shaft. We hypothesised that the negatively charged WT pilus (D69) is engaged electrostatically with

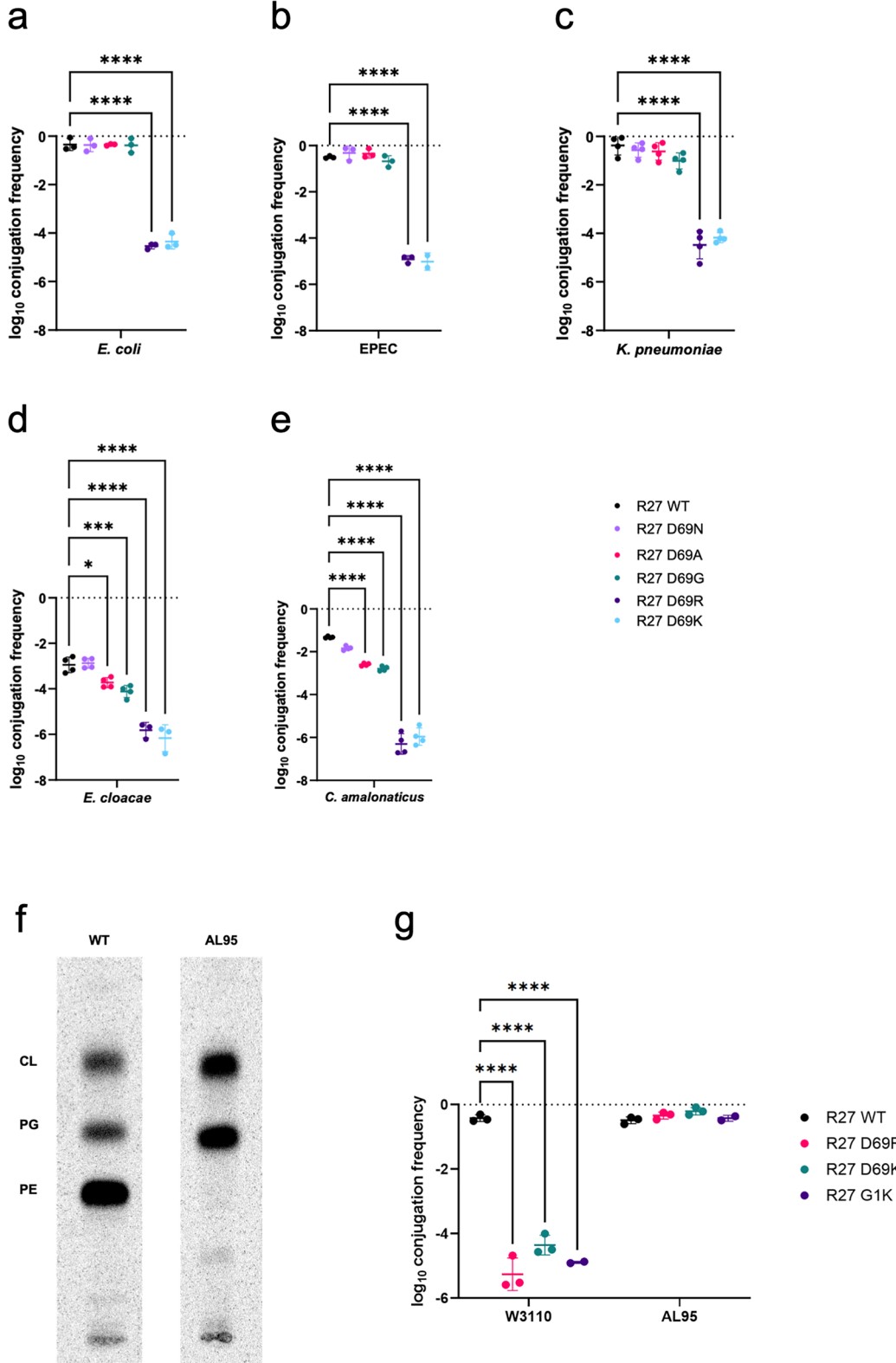

**Fig. 5 | Conjugation via TrhA WT, G1 and D69 substations into various recipients. a–e** Compared to WT TrhA, conjugation efficiency via the mutant TrhAs follows a gradient paralleling the sidechain size (N > A > G). Minimal conjugation was seen via TrhA D69R D69K. **f** Phospholipid profile of WT *E. coli* and PE-deficient mutant AL95. AL95 is PE-lacking due to the null allele of the *pssA* gene and contains mainly PG and CL. **g** Conjugation via via TrhA D69R, D69K and G1K mutants in restored into AL95 recipient. Data are shown as mean ± standard deviation. **a–e** $n$ = 2-4 biological replicates. Ordinary one-way ANOVA with Tukey's multiple comparison test, with a single pooled variance, *$p$ < 0.0332, ***$p$ < 0.0002, ****$p$ < 0.00001. **g** $n$ = 3 biological replicates. Ordinary two-way ANOVA with Šidák's multiple comparison test, with a single pooled variance. ****$p$ < 0.00001. Source data are provided as a Source Data file.

**Table 1 | Conjugation frequencies (Log$_{10}$ transformed) of non-polar D69 substitutions**

| TrhA | Ec | Ep | Kp | Ecl | Ca |
|------|-----|-----|-----|-----|-----|
| WT | −0.34 ± 0.25 | −0.5 ± 0.05 | −0.37 ± 0.38 | −2.94 ± 0.34 | −1.32 ± 0.04 |
| N | −0.36 ± 0.26 | −0.31 ± 0.3 | −0.56 ± 0.0.29 | −2.87 ± 0.21 | −1.83 ± 0.09 |
| A | −0.35 ± 0.03 | −0.34 ± 0.03 | −0.61 ± 0.36 | −3.72 ± 0.22 | −2.59 ± 0.06 |
| G | −0.37 ± 0.27 | −0.68 ± 0.29 | −1.01 ± 0.33 | −4.11 ± 0.26 | −2.78 ± 0.11 |
| R | −4.53 ± 0.11 | −4.91 ± 0.15 | −4.47 ± 0.57 | −5.37 ± 0.33 | −6.3 ± 0.47 |
| K | −4.35 ± 0.3 | −5.01 ± 0.37 | −4.71 ± 0.2 | −6.16 ± 0.59 | −5.96 ± 0.4 |

*Ec* E. coli, *Ep* EPEC, *Kp* K. pneumoniae, *Ecl* E. cloacae, *Ca* C. amalonaticus.

**Table 2 | Conjugation frequencies (Log$_{10}$ transformed) of polar substitutions biogenesis**

| TrhA | W3110 | AL95 |
|------|-------|------|
| WT | −0.4 ± 0.1 | −0.4 ± 0.1 |
| D69R | −5.26 ± 0.5 | −0.33 ± 0.1 |
| D69K | −4.35 ± 0.3 | −0.2 ± 0.11 |
| G1K | −5.16 ± 0.0.48 | −0.35 ± 0.12 |

zwitterionic PE in periplasmic leaflet of outer membrane. Consistently, the positively charged mutant pili (K1, K69 and R69) will be repelled from zwitterionic outer membrane of WT recipient but be attracted to the negatively charged outer membrane of the recipient, if such one exists. To test this hypothesis, we used WT *E. coli* and PE-deficient mutant, AL95, lacking phosphatidylserine synthase[31], as recipients. PE comprises about 75% of the phospholipid in the cell envelope of *E. coli* and other Gram-negative bacteria. The periplasmic (inner) leaflet of outer membrane contains normally up to 25-30% PE[24]. *E. coli* AL95 strain has no PE and contains only acidic phospholipids PG and cardiolipin (CL) (40 and 56% respectively) and phosphatidic acid, which altogether build a negatively charged inner and outer membrane, consisting 100% of negatively charged phospholipids. Our rationale was that in the absence of PE, but in the presence of negatively charged PG and cardiolipin the electronegative outer membrane would support conjugation of positively charged R27. Furthermore, we included R27$_{D69K}$, which also maintains the positive pilus charge but with a smaller sidechain. This showed that R27$_{D69R}$ and R27$_{D69K}$ as well as R27$_{G1K}$, were conjugated into AL95 in similar efficiency as R27$_{WT}$ is conjugated into WT *E. coli* due to electrostatic compatibility of the pilus and periplasmic (inner leaflet) of outer membrane of given recipient. These result support our hypothesis that electrostatic interactions between the positively charged R27$_{G1K}$, R27$_{D69R}$ and R27$_{D69K}$ with the negatively charged outer membrane of PE-deficient cells support an effective conjugation, but charge repelling interactions between positively charged R27$_{G1K}$, R27$_{D69R}$ or R27$_{D69K}$ and zwitterionic PE interfere with conjugation in this case, possibly due to interrupting the process of bringing the pilus tip into a conjugation permissive position for MPF and subsequent plasmid transfer. Accordingly, while the initial stages of pilus driven MPF formation is unclear, our observation suggests that the pilus surface charge and electrostatic compatibility play central role in MPF.

Taken together our data suggest that the amino acid composition of the pilus in the donor and the lipid composition of the outer membrane in the recipient might have mutually co-involved to ensure efficient conjugation. Remodelling of the Gram-negative bacterial outer membrane can affect conjugation primarily by altering the cell surface properties, which in turn modulates the efficiency of cell-to-cell contact and the transfer of genetic material. The structure and lipid composition of outer membrane of Gram-negative bacteria are regulated/tuned in response to environmental stress (temperature, pH, oxygen levels, nutrients, osmolarity and microenvironment of host tissues[36,37]. Therefore, adopting compositional and structural outer membrane changes would allow Gram negative bacteria to fine-tune conjugation efficiency in response to environmental cues, thereby increasing the chances of survival and the spread of beneficial genes, such as those conferring antibiotic resistance.

## Methods

### Identification and phylogenetic analysis of TrhA homologues

We first identified publicly available plasmids from the mating pair formation "F" (MPFF) group in the Plascad database[25]. We then used Plasmidfinder (database version 29/11/2021)[38] to identify the subset of these that carry IncH replicons. The TrhA protein sequence from the R27 plasmid (accession AF250878.1) was used as a query in a tBLASTn with BLAST+ v2.14.1[39] in order to identify homologues among the IncH plasmids. The resulting hits were aligned using MUSCLE v5.1.0[40]. RAxML-NG v1.2.0[41] was used to generate a maximum likelihood phylogenetic tree from the alignment. The phylogenetic tree was visualised together with metadata using Microreact[42].

### Site directed mutagenesis

The primers, plasmids and strains, used in this study are listed in Supplementary Tables 1-3. We previously constructed an open reading frame deletion of *trhA* on an R27 background (R27Δ*trhA*)[12]. Here we generated cloned the wild-type *trhA* gene into a mutagenesis plasmid (pSEVA612) and using site directed mutagenesis to change the D69 codon to that encoding N, A and R. These were inserted into the R27Δ*trhA* plasmid by homologous recombination[14]. Plasmids were constructed by Gibson Assembly (NEB) and a one-step KLD reaction used to introduce mutations. All mutations were confirmed by Sanger sequencing (Eurofins genomics).

### Bacterial conjugation

*E. coli* MG1655 Δ*trp* containing the derepressed *htdA* R27 mutant (drR27)[12], drR27*trhA*$_{D69A}$, drR27*trhA*$_{D69G}$, drR27*trhA*$_{D69N}$, drR27*trhA*$_{D69R}$, were used as donor; multiple bacterial species were used as recipients (Supplementary Table 3). The donor and recipients were grown overnight in LB at 37 °C, mixed in a ratio of 10:1, diluted 1 in 25 in PBS and 40 μL spotted onto LB agar plates (no selection). After incubation at 37 °C for 3 h, the plates were moved to 25 °C and incubated overnight. The conjugation mixture was then resuspended in 1 mL of PBS and serial dilution were plated on a M9 salts minimal media containing 4% glucose plates for selection of recipients and M9 salts minimal media containing 4% glucose and 30 μg/mL chloramphenicol selection plate for selection of transconjugants before incubating at 37 °C for 24 h. Data are presented as proportions of transconjugants/recipients. All conjugation experiments were done in three technical repeats and at least three biological repeats. The conjugation data were analysed by one-way Anova.

### Expression and purification of H-Pilus mutants

The H-pilus mutants were purified as we recently described[12]. Briefly, *E. coli* MG1655Δ*fimA* containing the R27 mutants was grown in 10 to

20 mL LB media with 10 µg/mL Chloramphenicol at 37 °C overnight. 10 mL were used to inoculate 1 L LB media, which was incubated with shaking overnight at 25 °C. The following steps were performed at 4 °C. The bacterial cells were harvested by centrifugation at 7000×g for 20 min and the pellet was resuspended in 20 mL PBS. The resuspended bacterial cells were shaved through 25 G needles 25 ~ 30 times and centrifuged at 50,000×g for 1 h. The supernatant was collected, and PEG 6000 was added at a final concentration of 5% with stirring for 1 h, which was followed by centrifugation at 50,000×g for 30 min. The supernatant was poured, and the pellet resuspended in 200 µL imaging buffer (50 mM Tris-HCL pH 8, 200 mM NaCl). The solution containing the filaments was dialyzed overnight in imaging buffer. Sample purity was judged by SDS-PAGE.

### Grid preparation and Cryo-EM data collection

Before preparing the cryoEM grid, the purified sample was centrifuged at 41,000 rpm for 30 min, and the pellet was resuspended with imaging buffer. 3 µL of the purified sample at ~4 mg/mL was applied to a glow-discharged carbon grid (Quantifoil R1.2/1.3, Cu, 300 mesh). The grid was blotted with a blotting force of 0 for 3 s at 4 °C and 100% humidity, and flash-frozen in liquid ethane using a Vitrobot Mark IV (Thermo Fisher Scientific). The frozen grids were stored in liquid nitrogen until data collection. Cryo-EM data were collected on 200 kV Glacios 2 (Thermo Fisher Scientific) with a Falcon 4 detector. A total of 200 – 711 movies were collected for each mutant dataset at a magnification of ×120,000 (pixel size of 1.192 Å/pixel) with a total electron dose of 46-48 e/Å2. The defocus range was set between −0.8 and −1.8 µm. All data were automatically acquired using EPU software. The data collection parameters are summarised in Supplementary Table 4.

### Cryo-EM data processing and model building

The data were processed using cryoSPARC (v.4.6.1)[43]. Each mutant dataset was collected and analysed independently. Every dataset was processed following similar steps (Supplementary Fig. 2). Motion correction was performed using Patch Motion Correction, and CTF estimation for micrographs was done by Patch CTF estimation[44]. Particles were automatically picked using a Filament tracer without templates and extracted in a binned state at 4.76 Å/pix. The extracted particles were classified in 2D, and selected high-quality particles were used as references for the Filament tracer. 2D classification of particles from each dataset was selected. Particles from a suitable 2D class were used for Helical refinement without helical parameters. Then, using the initial helical 3D map, the average power spectrum was calculated by helical symmetrical search. Symmetrical search job suggested an axial rise of ~12.2 Å and a twist of ~28.9°. These symmetry parameters were used as starting parameters for helical reconstruction using Helical refinement. The final H-pilus mutant helical reconstructions were shown 2.8-3.1 Å resolution with a rise of ~12.2 Å and a twist of ~29.0°.

The initial model of every mutant was based on PDBID: 9HVC. Real space refinement was performed in PHENIX[45]. The model was subjected to iterative cycles of refinement and manual rebuilding in COOT[46]. The structural model was validated using MolProbity[47]. The processing parameters and refinement statistics are summarized in in Supplementary Table Supplementary Table 4.

### Determination of steady-state phospholipid composition by radiolabeling

To determine the steady-state phospholipid composition, E. coli cells were uniformly labelled with 1 µCi/mL of [$^{32}$P]PO$_4$. After labelling, cells were harvested by centrifugation and resuspended in 0.2 mL of acidic solution (0.5 M NaCl in 0.1 N HCl). Lipids were extracted by adding 0.6 mL of solution 2 (chloroform:methanol, 1:2, v/v), followed by vortexing for 30 minutes. Subsequently, 0.2 mL of resuspension buffer was added, and the sample was vortexed for an additional 10 minutes.

Phase separation was achieved by centrifugation at 13,000×g for 5 minutes. The lower chloroform phase, containing the extracted phospholipids, was collected. An aliquot corresponding to approximately 1000–2000 counts per minute (cpm) of radiolabeled phospholipid was used for thin-layer chromatography (TLC) analysis. $^{32}$P-labeled phospholipids were resolved by one-dimensional thin layer chromatography (TLC) using HPTLC 60 silica gel plates (0.25 mm thickness) EMD, Gibbstown, NJ) activated and developed with a solvent consisting of chloroform:methanol:acetic acid (65:25:8 v/v). Radiolabeled lipids were detected using a Typhoon FLA 9500 phosphorimager and quantified with ImageQuant™ software. All TLC analyses were performed in at least two independent experiments to ensure reproducibility.

### Reporting summary

Further information on research design is available in the Nature Portfolio Reporting Summary linked to this article.

## Data availability

The cryo-EM maps have been deposited in the Electron Microscopy Data Bank (EMDB) under accession codes EMD-65235 (D69N), EMD-65234 (D69A), EMD-65250 (D69G) and EMD-65236 (D69R). The atomic coordinates have been deposited in the Protein Data Bank (PDB) under accession codes 9VP3 (D69N), 9VP2 (D69A), 9VPE (D69G) and 9VP4 (D69R). Raw movies for D69N, D69A, D69G and D69R were submitted to Electron Microscopy Public Image Archive (https://www.ebi.ac.uk/pdbe/emdb/empiar/) with IDs EMPIAR-13193, EMPIAR-13194, EMPIAR-13195 and EMPIAR-13196, respectively. The previously published 9HVC has been referenced. Source data are provided with this paper. The source data underlying Fig. 5, Tables 1 and 2 are provided as a Source Data file. Source data are provided with this paper.

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

## Acknowledgements

The cryo-EM experiments were performed at the UK National Electron Bio-Imaging Centre (eBIC) under visit NT29493 and the RIKEN Yokohama cryo-EM facility of the RIKEN Centre for Biosystems Dynamics Research (Yokohama). We would like to thank Ms Amika Kikuchi for her help with the cryo-EM data collection settings. This project made use the Computing Platform for Electron Microscopy at Imperial College funded by the BBSRC Mid-range equipment Initiative 22ALERT BB/X019284/1. SD is funded by the Bill & Melinda Gates Foundation (investment number INV-025280). This study was supported by a grant from The Wellcome Trust 224282/Z/21/Z to GF. This work was also supported by National Institutes of General Medical Sciences Grant R01GM121493-6 to MB. NI is funded by the Naito Foundation Fellowship, the grant for the 2025 Research Development Fund of Yokohama City University and by JSPS KAKENHI (JP25K18415).

## Author contributions

KB and GF conceived and supervised the study. SH performed conjugation assays. JLCW performed mutagenesis. JS-G performed bacterial cell maintenance. NI purified the RP4 pilus, determined the cryo-EM structure and performed model building and refinement. MB prepared the *E. coli* AL95 strain and analysed phospholipid composition. SD performed bioinformatic analysis. NI, GF and KB analyzed the structures. KB and GF wrote the manuscript with input from all authors.

## Competing interests

The authors declare no competing interests.
