## [Transparent Peer Review file · Nature Communications]

H pilin cyclisation and pilus biogenesis are promiscuous but electrostatic perturbations impair conjugation efficiency

Corresponding Author: Dr Konstantinos Beis

Version 0:

Reviewer comments:

Reviewer #1

(Remarks to the Author)

The manuscript follows up on the authors' recently published paper in PNAS describing a conjugative pilus derived from the IncH1 R27 plasmid. The latter was a striking finding in that it was the first structural demonstration of pilin cyclization through N-to-C terminal peptide bond formation. The latter had been proposed some years earlier but never unambiguously demonstrated through structural characterization. Notably, the authors have another paper in bioRxiv, cited in the manuscript, of a cyclic pilin in the RP4 system that, in addition, does not bind to lipid. These results are truly novel and of high impact. That being said, I struggle to find the main point of this manuscript. Three accomplishments stand out.

First, IncH replicons were identified and phylogenetically classified into 3 distinct sub-families, in which the members of the sub-families were associated with different groups of bacteria. Conserved sites were identified within the respective sequences. It was found that the Gly1 and Asp69 residues were absolutely conserved within the family.

Second a series of mutants was prepared in which D69 was substituted with G, A, N, and R. Cryo-EM reconstruction was employed to solve the structures of all 4 mutant pili at near-atomic resolution (reported resolutions from 2.82 - 3.13 Å). Given the reported resolutions, the structures are identical to the wt R27 pilus (PDB 9HVC). Clear density can be observed in the respective maps for the cyclic peptide bond. The data are of high quality. It is noteworthy that substitution of the conserved D69 residue has no obvious impact on the corresponding structure of either the pilus (same C5 helical symmetry with nearly identical values of rise and twist compared to the parent structure) or the cyclized TrhA protomer. One can view it as a negative result, noteworthy but providing limited insight into the reason that Asp69 is conserved. It does also raise a related point of whether substitutions were attempted at Gly1, which was not mentioned in the manuscript. By comparison, substitution of the N-terminal serine and the C-terminal glycine in the RP4 pilus showed some degree of promiscuity with regard to peptide cyclization, although apparently not to the same degree as the R27 pilus. However, it's difficult to infer too much information from this comparison due to small sample size, and differences in the mutations introduced and the identity of the C-terminal amino acid (Gly in RP4 and Asp in R27).

So, if substitution of Asp29 has no effect on pilin cyclization of pilus structure, does it have an effect on function to account for its conservation. The third set of experiments focuses on identification of a functional effect of substitution by looking at the dependence of conjugation efficiency with different mutants using different bacterial strains as recipients. However, the results are somewhat ambiguous. The D69R mutant had dramatically lowered conjugation efficiency in all assays that were performed. The authors attribute this effect to a potential interaction of the more positively charged outer surface of the pilus with the negatively charged bacterial surface. No experimental evidence is provided in support of the latter hypothesis. One challenge is that the surface receptor, or, more generally, host surface feature that interacts with the pilus, thereby promoting mating pair formation is not known—at least according to the authors. I accept their statement as I am not an expert in the field. The effect of the D69R mutation on conjugation efficiency could be due to charge reversal at a local interaction site. It could also be due to size as well. Mutants having smaller sized residues in place of D69 exhibit diminished conjugation efficiency. It's not unreasonable to speculate that the large size of the arginine sidechain could have a similar and perhaps more impactful effect on conjugation efficiency. Or, as they say, it could be something completely different with the point being that we don't know the answer and the manuscript doesn't provide it. Certainly the paper is solid experimentally and contributes meaningfully to the literature on bacterial conjugation systems. However, I don't see it as a high impact paper for the reasons summarized above.

Several additional comments:

1. The figures and tables in the SI pdf file are highly pixelated and difficult to read. Low resolution bitmap file with pdf conversion?
2. TrhA is transposed multiple times in the manuscript as ThrA-it's a bit confusing to the reader. There are other typos, e.g., Eentropathogenic in line 30, but I won't take the time summarize them
3. No discussion of modeling the lipid. In the original PNAS paper, the lipid was not unambiguously identified (even at the stated resolution of 2.2 Å), even though it was modeled as PG32:1. This should be briefly discussed in the methods.
4. Line 223: should read "N, A, G, and R"

Reviewer #2

(Remarks to the Author)

The manuscript describes the characterization of the TrhA pilin in the IncH plasmid R27, which polymerizes into the conjugative pilus and facilitates conjugation-mediated horizontal gene transfer. The TrhA pilin cyclizes via a peptide bond between Gly1 (G1) and Asp69 (D69) prior to association with a bound lipid (in a 1:1 pilin:lipid ratio) and assembly into the conjugative pilus. Given the rise and critical importance of antimicrobial resistance, and the important role that conjugation-mediated horizontal gene transfer of antimicrobial resistance genes (ARGs), understanding the structural consequences of changes to the pilin can affect contact-dependent gene transfer is important.

The authors analyzed the sequence diversity of TrhA pilins among 148 distinct plasmids; the G1/D69 pair is fully conserved in the TrhA pilins analyzed. Therefore, the authors probed the role of the second residue in the TrhA cyclization, D69, through a series of 4 mutants (D69N, D69A, D69G, and D69R), and how this residue can affect pilus structure, through a series of cryo-EM structures, and conjugation ability through mating assays. Structurally, all four TrhA mutants assembled pili, and the general parameters matched the WT pilus. While the lumen remained constant between the various mutants; the important changes noted were on the outer surface, particularly the charge distribution based the D69R mutation. Structural analysis of the mutants indicated that the D69R both alters the surface electrostatic potential but also sticks further out from the general surface of the pilus. It is this alteration of the surface electrostatic potential that apparently affects the ability of the D69R mutant to effectively function in conjugation. The authors posit that this alteration of surface charge disrupts a specific pilus – recipient receptor interaction (which remains elusively uncharacterized) and rather contributes to more non-specific membrane interactions between mutant donor and recipient. This disrupts effective mating pair formation and therefore loss of conjugative ability.

The manuscript is well constructed, direct, and efficient in the presentation of the data and the authors interpretations. The methods are clear, the mating assays are done with sufficient technical and biological replicates, though I wonder if the data presented in Table 1 is significant to 5 decimal places without any indication of standard deviations in the table; as I read the data in Table 1, 3 decimal places appear to be sufficient. Figure 5 does show what can be assumed to be an error bars, though the figure legend makes no note of it. The authors should clarify. Data collection and refinement of the cryo-EM structures are all reasonable as per their PDB EM validation reports. I would note that the supplementary materials, particularly Supplementary Figure 2 A-D and Supplementary Table S1 are not legible; the text is highly blurred. The authors should update this before publication.

There are also some more mundane, grammatical edits that the authors should address:

- Lines 123-124: The text "...cleaving the carboxy terminal five amino acids ..." reads better as "... cleaves the five carboxy terminal amino acids ..." (move "five")
- Line 125: change "involves" to "involving"
- Line 157: the text "TrbC G114C/L/T did not cyclase or polymerised into ..." should read "... TrbC G114C/L/T did not cyclize or polymerize into ..."
- Line 174: there is an extra "6" after 10⁶ in "10⁶ 6..." that should be removed.

Version 1:

Reviewer comments:

Reviewer #1

(Remarks to the Author)

My major concern with the original version of the manuscript was that the conclusions that were drawn were insufficiently supported by the experimental data. I don't believe that any experiments were in error, that is, the experimental data that were presented were sound. These considerations lowered my judgment of the potential impact of the manuscript. The investigators performed additional experiments that clarified some of the points that I had raised, in particular, as regards charge versus size effects with regard to the introduced mutations as well as the potential impact of the surface charge on conjugation efficiency. The experimental data support the conclusions drawn in the revised manuscript and heighten its potential impact. I have no reservations regarding the revised manuscript.

Reviewer #2

(Remarks to the Author)

The authors have addressed all my comments in their revised manuscript.

Response to reviewers' comments

We would like to thank both reviewers for their constructive comments which improved the paper. As a result of the new data generated in response to their suggestions, we modified the text and the title.

Reviewer #1

Comments: The comments ask: why D69 is conserved? is the defect in conjugation of the D69R due to size or charge? what are the consequences of a mutation in G1? As the useful comments made by the reviewer are interconnected, we provide a combined response.

Response: In the original submission we have shown that the pilus made by D69R was non-functional. We hypothesised that this was due to the lipid composition and charge of the outer membrane of the recipient. We have now confirmed this using a PE *E. coli* mutant as a recipient (new Fig. 5).

To address the reviewer question whether the effect on conjugation seen in D69R was due to charge or size, we generated a new mutant - D69K. As D69R and D69K behaved similarly in conjugation into WT and the PE mutant recipients, we ruled out size and concluded it was due to charge (new Fig. 5).

As suggested, we have generated a G1 substitution. As G1, like D69 is 100% conserved and is surface exposed, we made a G1K substitution. We show that pili made of G1K behave like those made from D69R and D69K (Fig. 5).

These data show that changing of either G1 or D69 could be detrimental and for the first time that the lipid composition of the outer membrane can affect conjugation outcomes.

Comment: Certainly the paper is solid experimentally and contributes meaningfully to the literature on bacterial conjugation systems. However, I don't see it as a high impact paper for the reasons summarized above.

Response: Thank you. We hope that with the new data generated in response to their comments the reviewer can now see the novelty and impact.

Several additional comments:

1. The figures and tables in the SI pdf file are highly pixilated and difficult to read. Low resolution bitmap file with pdf conversion?
2. TrhA is transposed multiple times in the manuscript as ThrA-it's a bit confusing to the reader. There are other typos, e.g., Eentropathogenic in line 30, but I won't take the time summarize them
3. No discussion of modeling the lipid. In the original PNAS paper, the lipid was not unambiguously identified (even at the stated resolution of 2.2 Å), even though it was modeled as PG32:1. This should be briefly discussed in the methods.

We have recently conclusively identified the R27 lipid. as PG, this data is integrated into the RP4 paper (doi: <https://doi.org/10.1101/2025.06.27.661960>).

4. Line 223: should read "N, A, G, and R"

Response: All the additional comments have been addressed.

Reviewer #2

Comment 1. I wonder if the data presented in Table 1 is significant to 5 decimal places without any indication of standard deviations in the table; as I read the data in Table 1, 3 decimal places appear to be sufficient.

Response: Modified

Comment 2. Figure 5 does show what can be assumed to be an error bars, though the figure legend makes no note of it. The authors should clarify.

Response: The legend has been corrected

Comment 3. I would note that the supplementary materials, particularly Supplementary Figure 2 A-D and Supplementary Table S1 are not legible; the text is highly blurred. The authors should update this before publication.

Response: Corrected

Comment: There are also some more mundane, grammatical edits that the authors should address:

- Lines 123-124: The text "...cleaving the carboxy terminal five amino acids ..." reads better as "... cleaves the five carboxy terminal amino acids ..." (move "five")

- Line 125: change "involves" to "involving"

- Line 157: the text "TrbC G114C/L/T did not cyclase or polymerised into ..." should read "... TrbC G114C/L/T did not cyclize or polymerize into ..."

- Line 174: there is an extra "6" after 10⁶ in "10⁶ 6..." that should be removed.

Response: Thank you, all have been corrected.